# Pre-Vaccination Immune Profiles and Responsiveness to Innate Stimuli Predict Reactogenicity and Antibody Magnitude Following mRNA Vaccination

**DOI:** 10.3390/vaccines13070718

**Published:** 2025-07-01

**Authors:** Amanda E. Zelkoski, Emilie Goguet, Emily Samuels Darcey, Mohamad-Gabriel Alameh, Hooda Said, Simon Pollett, John H. Powers, Eric D. Laing, Cara Olsen, Edward Mitre, Allison M. W. Malloy

**Affiliations:** 1Department of Pediatrics, Uniformed Services University of Health Sciences, Bethesda, MD 20814, USA; amanda.zelkoski.ctr@usuhs.edu; 2The Henry M. Jackson Foundation for the Advancement of Military Medicine, Bethesda, MD 20814, USA; emilie.goguet.ctr@usuhs.edu (E.G.); emily.samuels.ctr@usuhs.edu (E.S.D.); spollett@idcrp.org (S.P.); 3Department of Microbiology and Immunology, Uniformed Services University of Health Sciences, Bethesda, MD 20814, USA; eric.laing@usuhs.edu (E.D.L.); edward.mitre@usuhs.edu (E.M.); 4Department of Pathology and Laboratory Medicine, Children’s Hospital of Philadelphia, Philadelphia, PA 19104, USA; mg.alameh@pennmedicine.upenn.edu (M.-G.A.); saidh@chop.edu (H.S.); 5Department of Pathology and Laboratory Medicine, University of Pennsylvania, Philadelphia, PA 19104, USA; 6Center for Cellular and Molecular Therapeutics, Children’s Hospital of Philadelphia, Philadelphia, PA 19104, USA; 7Infectious Disease Clinical Research Program, Department of Preventive Medicine & Biostatistics, Uniformed Services University of the Health Sciences, Bethesda, MD 20814, USA; 8Clinical Research Directorate, Leidos Biomedical Research, Inc., Frederick National Laboratory for Cancer Research, Frederick, MD 21701, USA; powersjohn@mail.nih.gov; 9Department of Preventive Medicine & Biostatistics, Uniformed Services University of the Health Sciences, Bethesda, MD 20814, USA; cara.olsen@usuhs.edu

**Keywords:** mRNA vaccine, BNT162b2, antibody, dendritic cells, monocytes, reactogenicity, correlates, predictors, innate immune response

## Abstract

Background: While mRNA vaccines effectively limit hospitalization and severe COVID-19 disease, the precise early innate immune mechanisms associated with their efficacy and reactogenicity remain underexplored. The identification of innate immune correlates prior to vaccination could provide mechanistic insights and potentially predict responses. Methods: We developed an in vitro model to study the innate immune activation of pre-vaccination peripheral blood mononuclear cells (PBMCs) collected from participants enrolled in a well-characterized COVID-19 BioNTech/Pfizer BNT162b2 vaccine (BNT162b2 vaccine) cohort. Pre-vaccination PBMCs were stimulated with empty lipid nanoparticle (LNP), mRNA-LNP, or Toll-like receptor (TLR) agonists. Using multiparameter spectral flow cytometry, we analyzed the baseline immune state, innate responsiveness to stimuli, and cytokine profiles of study participants. These pre-vaccination in vitro results were analyzed for correlations with post-vaccination symptoms and spike-specific IgG responses. Results: Baseline dendritic cell (DC) states inversely correlated with the magnitude of symptoms following BNT162b2 vaccination. Heightened conventional (cDC) and weaker plasmacytoid DC (pDC) responses to RNA stimuli correlated with the magnitude of an acute IgG response. IgG durability modestly correlated with a lower pDC state but higher cDC2 and monocyte baseline states and inversely correlated with TLR3 agonist responsiveness. Conclusions: The pre-vaccination assessment of innate immune function and resting states can be used to fit models potentially predictive of immunogenicity and reactogenicity to BNT162b2 vaccination. Pre-vaccination DC states may influence reactogenicity, while the response to RNA may impact antibody responses. Our data suggest that pre-vaccination assessment offers insights into the innate mechanisms driving mRNA vaccine responses and has predictive potential.

## 1. Introduction

mRNA vaccines are highly effective at producing protective immune responses against severe and long COVID-19 [1,2,3,4]. The mRNA platform has enabled reduced time to development and production compared to prior vaccine strategies, making it a highly relevant and useful tool in the prevention of disease. While the magnitude of antibody responses to vaccine antigens, particularly neutralizing antibodies, has been identified as a correlate of protection [5], variability exists in the magnitude of spike-specific antibody responses [5,6,7] and the severity of local and systemic side effects experienced by individuals [6]. This suggests that the innate immune response to mRNA vaccines may influence these outcomes. Furthermore, the durability of the response to mRNA vaccination in general wanes within 6 months of administration [8,9].

The mechanisms by which mRNA vaccines induce humoral and cellular immunity have been investigated. Initial studies showed that cytokine profiles for interferon gamma (IFN-γ) and CXCL10 were upregulated, along with circulating monocyte subsets in the peripheral blood one day after vaccination [10]. Building upon these findings, additional groups evaluated cytokines and innate immune cell activation within the first two days post-vaccination, aiming to correlate them with antibody magnitude and vaccine-associated symptoms [11,12]. One study identified a correlation between a reduction in peripheral blood CD4+ T cell counts after the first vaccine dose and increased antibody magnitude [12]. Furthermore, reductions in NK cells and non-classical monocyte frequencies post-second dose were associated with greater neutralizing antibody titers [12]. These collective findings underscore the roles of early innate immune activation and recruitment as potential correlates of immunogenicity.

Reactogenicity primarily arises from the innate inflammatory response. This response leads to the production of pro-inflammatory cytokines such as IL-6, IL-1β, and TNFα, which mediate inflammation and pain at the injection site [13,14,15]. Upon entering circulation, these cytokines contribute to systemic symptoms like fever and fatigue [14]. In individuals receiving COVID-19 mRNA vaccines, increased levels of innate cytokines and chemokines such as CXCL9, CXCL10, CXCL11, IFN-γ, and CCL20 have been identified in the peripheral blood of adults, with CCL20 and IFN-γ showing the strongest correlation with post-vaccination symptoms [11].

The relationship between innate immune activation associated with reactogenicity and the induction of adaptive immunity remains unclear. Some studies have reported that pre-vaccination inflammatory gene signatures correlate with both reactogenicity and robust adaptive immune responses [13,16,17,18]. However, other investigations challenge this connection, finding no association between reactogenicity and the strength of vaccine-induced adaptive immunity [19,20]. These divergent findings suggest that different patterns of pre-vaccination immunophenotypes may be associated with immunogenicity and reactogenicity. This understanding could inform the development of vaccines designed to elicit robust immune responses while minimizing adverse reactions.

Ideally, in vitro predictors of immunogenicity and reactogenicity could be identified prior to vaccination. Previous studies evaluating pre-vaccination innate immune profiles have identified patterns associated with antibody titers elicited by other vaccines. For instance, the pre-vaccination baseline expression of genes involved in antigen presentation, B cell receptor (BCR) and pattern recognition receptor (PRR) signaling, apoptosis, and inflammation has been linked to immune responses against influenza, hepatitis B (HBV), and yellow fever vaccines [21,22,23,24,25]. Furthermore, the upregulation of type I interferon (IFN) response pathways and Toll-like receptor (TLR) 7/8 signaling following vaccine stimulation has been associated with enhanced adaptive responses to influenza vaccination [18,26]. While the innate correlates of adaptive immune responses vary, some studies indicate that baseline innate immune factors correlate with the induction of robust adaptive responses across vaccine platforms [15,27]. However, to date, pre-vaccination innate correlates for mRNA vaccination outcomes have not been identified.

The objective of this study was to gain insights into mRNA vaccine-induced outcomes by investigating the predictive potential of baseline immune and stimuli-induced innate activation for both reactogenicity and immunogenicity. Furthermore, we aimed to fit predictive models to potentially identify individuals predisposed to increased symptom severity or a poor antibody response. To accomplish this, we stimulated pre-vaccination peripheral blood mononuclear cells (PBMCs) in vitro with mRNA-LNP, empty LNP, and TLR agonists. Our analysis reveals distinct baseline innate immune states and innate immune response profiles that correlate with reactogenicity and antibody responses at both 1- and 6-months post-vaccination.

## 2. Materials and Methods

### 2.1. Study Participants

Study participants were enrolled in the Prospective Analysis of a SARS-CoV-2 Seroconversion (PASS) observational cohort [28]. The PASS cohort consisted of 271 healthcare workers who were ≥18 years old, not severely immunocompromised, and seronegative for SARS-CoV-2 upon enrollment between August 2020 and March 2021. PASS participants were followed up monthly for the first year and quarterly for the second year at the Naval Medical Research Center (NMRC) Clinical Trials Center or the Uniformed Services University (USU) Translational Medicine Unit. Across serial follow-up visits, blood was drawn, PBMCs were collected and cryopreserved, and sera were stored for antibody testing.

The subset of participants included for analysis in this study met the following additional criteria: (1) received the FDA-approved 2-dose COVID-19 BioNTech/Pfizer BNT162b2 mRNA vaccine (BNT162b2 vaccine), (2) completed post-vaccination symptom questionnaires, (3) had serum samples from 1- and 6-months post vaccination time points, (4) had cryopreserved pre-vaccination PBMCs with viability >70% after thawing, and (5) had anti-spike binding IgG antibody serum levels in the upper or lower quintile 1 month after the 2nd vaccination. While participants were initially selected based on anti-spike binding IgG antibody serum levels falling into the upper or lower quintile one month after the second vaccination to ensure a spread of antibody responses, the separation between these two populations proved minimal. Consequently, these participants were treated as a single cohort rather than distinct high- and low-responder groups. Twenty-nine participants met these criteria and were included in this analysis.

This PASS study protocol was approved by the USU Institutional Review Board, and participants provided written informed consent prior to enrollment.

### 2.2. Antibody Testing

Sera were collected monthly during the first year following study enrollment and tested for IgG antibodies against the Wuhan-1 SARS-CoV-2 spike protein as previously described [9]. Briefly, binding IgG antibodies against spike protein ectodomain trimers were measured using a microsphere-based multiplex immunoassay (MMIA). Anti-spike IgG levels were measured on Bio-Plex 200 multiplexing systems and were quantified in WHO binding antibody units per mL (BAU/mL) by interpolation against an internal reference standard that had been calibrated against a US Human SARS-CoV-2 Serology Standard provided by the Frederick National Laboratory. Anti-SARS-CoV-2 IgG levels measured at 1 month (range: 21–44 days) and 6 months (range: 166–200 days) post-vaccination with the BNT162b2 vaccine were published previously [9].

### 2.3. Post-Vaccine Symptom Assessment

In the first visit after each dose of the BNT162b2 vaccine, participants completed a vaccine-associated symptoms questionnaire as previously described [28,29]. In brief, participants were surveyed regarding the presence and severity of 8 systemic, 3 local symptoms, 1 non-local and non-systemic symptom. The systemic symptoms measured were feeling weak or tired, having a headache, body aches or pain, joint pain, nausea, feeling hot, feeling cold, and having chills or shivering (Table A1). The local symptoms evaluated were soreness, pain, or redness at the injection site. The non-local and non-systemic symptom asked about on the questionnaire was lymph node swelling. Individuals were asked to rank the intensity of each symptom on a scale of 0–4, with 0 indicating the absence of symptoms and 4 being the most intense. Scores were summed for a symptom severity score (range: 0–48).

### 2.4. PBMC Isolation and Purification

PBMCs were isolated and cryopreserved from PASS participants at enrollment and subsequent visits as previously described [28,30].

### 2.5. Formulation of LNP and mRNA-LNP

Empty LNPs and mRNA-LNPs were synthesized as follows. The empty LNP formulation included ALC-0315 ionizable lipid, cholesterol, DSPC, and the ALC-0159 PEG-lipid (molar ratio: 46.3:42.7:9.4:1.6), consistent with the BNT162b2 LNP. The mRNA encoding codon-optimized, SARS-CoV-2 Omicron variant di-proline-modified spike glycoprotein (S2P) was encapsulated by LNPs via rapid microfluidic mixing as previously described [31]. These LNPs were produced and characterized by the Alameh lab. Hydrodynamic size, polydispersity index (PDI), and surface charge (ζ-potential) were determined by dynamic light scattering (DLS) and Laser Doppler Velocimetry using a Malvern Zetasizer Ultra (Red Label). Encapsulation efficiency and concentration for the mRNA-LNP were assessed via the RiboGreen RNA assay (Invitrogen, Waltham, MA, USA). The empty LNP exhibited a hydrodynamic size of 54.07 nm, a PDI of 0.28, and a zeta potential of −4.447 mV. The mRNA-LNP had an encapsulation efficiency of 98.17%, a 66.4 nm diameter, a 0.1090 PDI, and a zeta potential of −11.55 mV. To facilitate direct comparison with the mRNA-LNP, the concentration of the empty LNP is presented as the equivalent mRNA concentration and has the same lipid concentration as determined by HPLC-CAD.

### 2.6. Cell Preparation and Stimulation

PBMCs collected at enrollment were rapidly thawed in a 37 °C water bath and transferred to a 15 mL conical tube containing pre-warmed R10 media. R10 media consisted of RPMI 1640 with 2 mM L-glutamine (Cytiva, Marlborough, MA, USA), supplemented with 10% heat-inactivated FBS, and Pen–Strep (100 U/mL–100 µg/mL). Following a wash, cells were resuspended in R10 containing 30 μg/mL of DNase I (Roche, Basel, Switzerland) and rested at 37 °C and 5% CO_2_ for 45 min.

For each participant, 1 × 10^6^ cells were immediately stained for flow cytometry in the absence of in vitro incubation or stimulation and are referred to in the text as baseline.

The remaining cells were washed, resuspended at 5 × 10^6^ cells/mL, and aliquoted at 1 × 10^6^ cells per well of a polystyrene 24-well plate (Cat: 3526, Corning Inc., Corning, NY, USA). Stimulation was added to each well: VacciGrade R848 (0.25 µg/mL) (Invivogen, San Diego, CA, USA), VacciGrade Poly I:C (25 µg/mL) (Invivogen, San Diego, CA, USA), synthetic Monophosphoryl lipid A (MPLA) (0.25 µg/mL) (Invivogen, San Diego, CA, USA), an empty LNP (1 µg/mL), or mRNA-LNP (1 µg/mL). A separate well served as an unstimulated control.

After adding stimuli, the volume of the wells increased to 1 mL with R10 and was mixed thoroughly. The plate was then incubated at 37 °C and 5% CO_2_ for 24 h. Post-incubation, cells were transferred to 1.5 mL tubes and pelleted. Supernatant was collected and stored at −80 °C for cytokine analysis. Cell pellets were washed with ice-cold PBS-2% FBS and stained for flow cytometry.

Doses were determined based on the titration analysis of the upregulation of costimulatory molecules, CD80 and CD86, on monocytes (Figure A1). When optimal doses conflicted for costimulatory molecule expression, the expression of CD86 was used to determine the dose, as it was the most consistent across doses. We confirmed no significant differences in viability between stimulation conditions: unstimulated at 88.4% (70.2–97.7%), empty LNP at 88.5% (71.7–95.4%), mRNA-LNP at 88.5% (70.6–95.5%), MPLA at 88.4% (77.3–96.9%), Poly I:C at 87.0% (74.5–96.4%), and R848 at 89.3% (71.6–97.1%).

### 2.7. Flow Cytometry

To prepare PBMCs for flow cytometry, cells were washed with PBS-2% FBS and incubated with Human TruStain FcX (BioLegend, San Diego, CA, USA) at 4 °C for 10 min to inhibit the nonspecific binding of antibodies. Cells were subsequently incubated with a surface-staining flow cytometric antibody mixture (Table A2) at 4 °C for 20 min. After an additional wash with PBS-2% FBS, cells were fixed with BD Cytofix buffer (BD Biosciences, Franklin Lakes, NJ, USA) at 4 °C for 30 min. The fixed cells were resuspended in a 1:1 mixture of BD Cytofix buffer and PBS-2% FBS and transferred to FACS tubes for acquisition. Data were collected using a Cytek Aurora Spectral Cytometer (Cytek Biosciences, Fremont, CA, USA) and analyzed using FlowJo v10 (BD Biosciences, Franklin Lakes, NJ, USA).

The immune cells characterized in this study were gated as shown in Figure A2A. Innate immune cell subsets, including monocytes, plasmacytoid dendritic cells (pDCs), and both conventional DC types 1 and 2 (cDC1, cDC2), were identified by sequential gating. Initial gates removed debris, doublets, and dead cells. Subsequently, T cells (CD3^+^ CD11c^−^ HLA-DR^−^) and B cells (CD19^+^ CD11c^−^ HLA-DR^+^) were identified. Within the remaining population, natural killer (NK) cells were defined as CD56^+^ HLA-DR^−^ (Figure A2A). Within the HLA-DR^+^ population, monocytes were identified by the expression of CD14^+^ and/or CD16^+^. From the CD14^−/lo^ and CD16^−^ non-monocyte population, pDCs were identified as CD123^+^ CD11c^−^. Among CD11c^+^ cells, cDC2s were defined as HLA-DR^hi^ CD1c^+^, and cDC1s as HLA-DR^hi^ CD141^+^ (Figure A2A). Due to the low frequency of cDC1s, a broad gate was employed to ensure the comprehensive capture of this population.

### 2.8. Cytokine Analysis

To quantify cytokine production, the LEGENDplex™ Human Anti-Virus Response Panel 1 V02 (IFN-λ1(IL-29), IL-1β, IL-6, TNFα, IP-10, IL-8, IL-12p70, IFN-α2, IFN-λ2 (IL-28A), GM-CSF, IFN-β, IL-10, IFN-γ) (BioLegend, San Diego, CA, USA) was employed using the supernatant of incubated PBMCs. The assay was conducted following the manufacturer’s protocol. Briefly, frozen supernatants were thawed on ice and diluted using the provided assay buffer. Diluted samples and standards were plated in duplicate into a 96-well V-bottom plate. After the addition of Human Anti-Virus Response capture beads, samples were incubated at room temperature for 2 h in the dark with shaking. Following this incubation, the plate was washed with the designated wash buffer, and detection antibodies were added. After a 1 h room temperature incubation in the dark with shaking, streptavidin–PE was added without an intermediate wash. A final 30 min incubation was immediately followed by an additional wash. Wells were resuspended in additional wash buffer, and samples were acquired on the Cytek Aurora Spectral Cytometer (Cytek Biosciences, Fremont, CA, USA). Data were analyzed using the cloud-based Qognit data analysis software (Qognit, San Carlos, CA, USA). The cytokine concentrations were determined by the software through the generation of a standard curve.

### 2.9. Statistical Analysis

For correlative analysis, we hypothesized a priori that the magnitude of monocyte and cDC activation by empty LNPs and MPLA, a TLR4 agonist, correlated with the IgG response at both 1- and 6-months post-vaccination. The level of activation was quantified as the change in CD80 and CD86 expression compared to the unstimulated control. Additionally, we hypothesized that the IL-6, TNF-α, and IL-1β produced by PBMCs stimulated with the empty LNP and MPLA correlated with the symptom score. Secondary analyses were performed incorporating the additional measures obtained in this study. These additional measures included the baseline expression of activation markers, as well as a change in the expression of activation markers and the production of cytokines in response to stimuli. Significance was determined as *p* < 0.05. No corrections for multiple comparisons were performed, as secondary analyses were designed to be exploratory. Normality tests were run prior to each comparison. As the distribution of the data was non-parametric, Spearman correlations were performed, and the presence of outliers was assessed. The correlation coefficient for Spearman correlations is given where applicable. Spearman correlations were performed using GraphPad PRISM v10 (GraphPad Software Inc., Boston, MA, USA).

Stepwise multiple linear regressions, employing both forward and backward selection procedures based on the Akaike Information Criterion (AIC), were performed using R (v. 4.4.1) with the MASS package (v. 7.3) [32]. Multiple linear regression analyses were performed with (1) surface marker expression (mean fluorescence intensity [MFI] or ΔMFI) within a cell population, (2) surface marker expression (MFI or ΔMFI) across all cell populations, (3) the production of all cytokines, and (4) surface marker expression (ΔMFI) across all cell populations with cytokine production. These analyses were conducted separately for each condition: empty LNP stimulation, mRNA-LNP stimulation, MPLA stimulation, Poly I:C stimulation, R848 stimulation, and baseline. Baseline, also referred to as the “resting state,” is defined as the activation marker expression evaluated immediately post-thaw (MFI). Multiple R-squared values were reported in the text, accompanied by the respective *p* values. Only statistically significant comparisons were reported.

## 3. Results

### 3.1. Analysis of Immunogenicity and Reactogenicity in Recipients of BNT162b2 Vaccination

In this study, we sought to define innate immune activation signatures predictive of reactogenicity or immunogenicity. We analyzed PBMCs from the selected participants who were predominantly female (82.7%) and white (65.5%) with an age range of 33–51 years and a median age of 41. Anti-SARS-CoV-2 IgG levels were measured at 1-month post-vaccination with the BNT162b2 vaccine (range: 21–44 days) and at 6 months (range: 166–200 days) [9]. Reactogenicity to the BNT162b2 vaccine was assessed via a symptom questionnaire post-boost, and symptom scores ranged 0–38 (Table 1 and Table A1) [7]. Consistent with the findings of the entire PASS cohort [7], the magnitude of anti-spike IgG antibodies at 1- and 6-months post-boost did not correlate with the reactogenicity score on the post-boost symptoms questionnaire (Figure 1A).

We hypothesized that pre-vaccination innate immune responses, mediated by RNA or LNP-sensing signaling pathways, would correlate with the magnitude of anti-spike IgG responses or vaccine-associated symptoms after the second dose of the primary vaccine series. We therefore stimulated pre-vaccination PBMCs with BNT162b2 vaccine components, including empty LNPs and LNPs containing SARS-CoV-2 spike-encoding modified mRNA (mRNA-LNP). Additionally, we stimulated the PBMCs with TLR agonists targeting RNA- or LNP-sensing pathways: MPLA (TLR4), Poly I:C (TLR3), and R848 (TLR7/8). Poly I:C targets TLR3, an endosomal receptor known to detect both mRNA secondary structures and double-stranded RNA, which can be byproducts of mRNA synthesis, potentially present as a residual component of vaccine production [33]. The vaccine mRNA has been modified to evade TLR7/8 detection, which is an endosomal TLR that recognizes single-stranded RNA [34]. R848 is a small molecule that targets TLR7/8. Several studies indicate that LNPs are capable of cellular activation through TLR4 [35,36].

Baseline expression as well as stimulation-induced change in activation marker expression were correlated with symptom scores following the second dose of the BNT162b2 vaccine using Spearman correlations and multivariate linear regression.

### 3.2. Baseline Innate Immune Phenotypes Associated with Reactogenicity

Innate immune activation plays a crucial role in vaccine reactogenicity, with activated monocytes, DCs, and NK cells producing inflammatory cytokines that contribute to adverse reactions [14]. We thus investigated the relationship between the cellular responsiveness to innate immune ligands present in mRNA vaccines and vaccine reactogenicity.

Our analysis highlights the expression of multiple activation markers on monocytes as correlates of vaccine reactogenicity. In the heatmap shown in Figure 1B, dark blue indicates a strong positive correlation, dark red indicates a strong negative correlation, and white indicates no correlation. The stimulation of PBMCs with empty LNPs revealed that the upregulation of CD86 on monocytes correlated with higher symptom scores (rho = 0.445, *p* = 0.016), while the upregulation of CD25 correlated with lower scores (rho = −0.4857, *p* = 0.008, Figure 1B). Similarly, mRNA-LNP-induced upregulation on the monocytes of CD86 (rho = 0.5522, *p* = 0.002), HLA-DR (rho = 0.4776, *p* = 0.009), and CD14 (rho = 0.3928, *p* = 0.035) were each significantly associated with reactogenicity. As with empty LNP stimulation, the increased expression of CD25 on monocytes in response to mRNA-LNP was inversely associated with reactogenicity (rho = −0.442, *p* = 0.164, Figure 1B). Monocyte activation by certain TLR agonists significantly correlated with vaccine reactogenicity symptom scores (Figure 1B). Specifically, MPLA- and R848-induced CD40 expression showed positive correlations (rho = 0.4724, *p* = 0.010 and rho = 0.3788, *p* = 0.043, respectively), as did R848-induced CD70 expression (rho = 0.4257, *p* = 0.021).

B cell responses also correlated with symptom scores. The LNP-induced upregulation of HLA-DR on B cells showed a positive association with reactogenicity (rho = 0.377, *p* = 0.044), with a similar, though not statistically significant trend observed for mRNA-LNP. The R848 activation of B cells revealed positive associations with reactogenicity for CD86 (rho = 0.3847, *p* = 0.039), CD40 (rho = 0.4195, *p* = 0.024), and CD25 (rho = 0.3739, *p* = 0.046, Figure 1B). Similarly, the Poly I:C treatment-induced expression of CD40 (rho = 0.4363, *p* = 0.018), CD70 (rho = 0.328, *p* = 0.40), and CD80 (rho = 0.498, *p* = 0.006) on B cells aligned with the magnitude of the symptom scores recorded by study participants (Figure 1B).

NK cell activation in response to innate ligands also correlated with reactogenicity scores. The NK cell expression of CD38, a marker of NK cell activation [37], was consistently associated with symptom scores across most tested conditions (Figure 1B). Positive correlations were observed with responses to the empty LNP (rho = 0.4091, *p* = 0.028), mRNA-LNP (rho = 0.5493, *p* = 0.002), MPLA (rho = 0.4615, *p* = 0.012), and Poly I:C (rho = 0.418, *p* = 0.024). These findings align with prior studies linking NK cell responses to mRNA vaccine reactogenicity [12,30].

The baseline expression of activation markers was inversely correlated with reactogenicity, particularly in cDC2 and pDC populations (Figure 1B). The lower baseline expression of CD70 on monocytes (rho = −0.3844, *p* = 0.040); CD70 (rho = −0.3839, *p* = 0.040) and CD25 (rho = −0.4274, *p* = 0.21) on cDC2s; CD86 on cDC1s (rho = −0.5707, *p* = 0.001); and CD80 (rho = −0.3783, *p* = 0.043), CD70 (rho = −0.5618, *p* = 0.002), CD25 (rho = −0.5122, *p* = 0.005), and HLA-DR (rho = −0.4969, *p* = 0.006) on pDCs all associated with greater reactogenicity (Figure 1B).

Following a univariate analysis, we used multiple linear regression to define clusters of activation markers within each immune cell population that exhibited the strongest association with reactogenicity. Monocyte activation following mRNA-LNP stimulation exhibited a significant association with symptom variance, accounting for 56.7% of the observed variability in the post-boost symptom score (multiple R^2^ = 0.567, *p* = 0.001) (Table 2). B cell activation by Poly I:C stimulation, as defined by CD80 and CD40 expression, accounted for 43.32% of symptom variance (multiple R^2^ = 0.4332, *p* ≤ 0.001).

Furthermore, as there were significant associations between CD38 upregulation on NK cells and reactogenicity (Figure 1B), we assessed NK activation measured by CD38 in combination with CD70, CD25, and CXCR4. The combined upregulation of CD38 and these additional markers explained 20–34% of symptom variance across various stimuli (mRNA-LNP: multiple R^2^ = 0.3233, *p* = 0.006; MPLA: multiple R^2^ = 0.206, *p* = 0.050; Poly I:C: multiple R^2^ = 0.3582, *p* = 0.003; R848: multiple R^2^ = 0.3448, *p* = 0.004, Table 2). Lastly, baseline activation marker expression showed a strong negative association with reactogenicity, with pDC costimulatory molecule expression at baseline accounting for 47.95% of the variance in post-boost symptom scores (multiple R^2^ = 0.4795, *p* = 0.001, Figure 1B, Table 2).

Expanding on our findings, we next used multiple linear regression to define clusters of activation markers across each immune cell population that exhibited the strongest association with reactogenicity. The strongest association was found with baseline immune states. Our baseline expression model, which included markers from pDCs (CD70 and CD25), B cells (CD40, HLA-DR, and CD38), cDC1s (CD86), and cDC2s (HLA-DR and CXCR4), explained 78.95% of the variability in experienced symptoms (multiple R^2^ = 0.7895, *p* ≤ 0.001) (Table 2). This result suggests that a relatively higher baseline state of activation or “response readiness” is associated with reduced reactogenicity to the BNT162b2 vaccine.

Beyond baseline, we also identified combinations of activation markers across cell populations that were significantly associated with symptom scores. The strongest models for each condition had the multiple R^2^ values of 0.2862 (empty LNP), 0.5446 (mRNA-LNP), 0.5078 (MPLA), 0.5753 (Poly I:C), and 0.2902 (R848) (Table 2). The robust associations seen with the mRNA-LNP, MPLA, and Poly I:C models indicate that interparticipant variability in RNA and LNP sensing and signaling may influence mRNA vaccination-associated symptoms. These findings collectively underscore the ability of baseline immune readiness and cellular responsiveness to develop potentially predictive models for individual reactogenicity profiles to BNT162b2.

### 3.3. Multiple Regression Modeling Identified Associations Between Cytokine Responses to RNA Sensor Stimulation and Reactogenicity

Reactogenicity has been associated with the production of cytokines and chemokines in human in vivo studies and in animal models [13,14,19,36]. Therefore, we hypothesized that proinflammatory cytokine production upon the stimulation of pre-vaccination PBMCs with mRNA-LNPs, empty LNPs, or TLR agonists could identify cytokine responses correlating with BNT162b2 vaccine reactogenicity. An analysis of cytokine production by stimulated PBMCs revealed positive associations between the magnitude of cytokines produced and symptom scores (Figure 1C). Among the tested conditions, R848-induced IFN-β production (rho = 0.3746, *p* = 0.045) exhibited a statistically significant correlation (Figure 1C).

As with surface marker analysis, we used multiple linear regression to identify cytokine combinations that more strongly correlated with the degree of symptoms experienced after mRNA vaccination. This analysis revealed that the mRNA-LNP-induced production of IP-10, IL-8, and IFN-γ accounted for 48% of the variance (multiple R^2^ = 0.4834, *p* ≤ 0.001) in symptom scores between participants. Similarly, Poly I:C-induced IFN-β, IL-8, and GM-CSF explained 45% of symptom score variability (multiple R^2^ = 0.4504, *p* = 0.002) (Table 2).

Following the analysis of cytokine combinations, we performed multiple linear regression to assess if their incorporation into our surface marker models would strengthen the correlation with post-boost symptoms. This integration successfully enhanced the models for mRNA-LNP (multiple R^2^ = 0.5875, *p* ≤ 0.001) and Poly I:C (multiple R^2^ = 0.6347, *p* ≤ 0.001) stimulation (Table 2). These findings demonstrate that cytokine profiles, particularly those linked to RNA sensor activation, correlate with mRNA vaccine reactogenicity.

### 3.4. Activation of cDC2 and pDC Populations Correlate with Antibody Response 1-Month Post-Vaccination

We next sought to identify distinct cellular and cytokine profiles associated with SARS-CoV-2 anti-spike IgG magnitude. Participants in this study were selected to have demonstrated antibody response to vaccination and exhibit a wide variance in anti-spike IgG levels (Figure 2A). We hypothesized that mRNA-LNP-induced innate activation would demonstrate the strongest correlation with the magnitude and durability of antibody responses.

The mRNA-LNP stimulation of pre-vaccination PBMCs from study participants revealed correlations with 1-month antibody titers in monocytes, cDC2, and pDCs. In monocytes, mRNA-LNP-induced CD86 expression (rho = 0.3883, *p* = 0.037) significantly correlated with IgG levels (Figure 2B). In cDC2s, mRNA-LNP-induced CD40 upregulation was associated with 1-month titers (r = 0.3749, *p* = 0.045). Conversely, in pDCs, the mRNA-LNP-induced upregulation of CD70 was inversely correlated with antibody levels (rho = −0.4067, *p* = 0.029), as was CD25 upregulation (rho = −0.384, *p* = 0.040).

Across TLR stimulation conditions, distinct patterns of activation correlated with the anti-spike IgG response magnitude. The B cell expression of CD38 after stimulation with Poly I:C exhibited a statistically significant correlation with the antibody response (rho = 0.4034, *p* = 0.030, Figure 2B).

cDC1 and pDC populations displayed divergent associations with 1-month antibody levels. R848-stimulated cDC1s showed positive correlations with antibody levels, particularly for CD80 (rho = 0.5291, *p* = 0.003) and CD40 (rho = 0.4084, *p* = 0.028, Figure 2B). In contrast, the MPLA stimulation of cDC1s resulted in a significant inverse correlation between CD86 expression and IgG levels (rho = −0.5108, *p* = 0.005). For pDCs, increased CD70 expression following MPLA (rho = −0.4404, *p* = 0.017) and R848 (rho = −0.5365, *p* = 0.003) treatment significantly correlated with lower titers. This inverse relationship was also seen with CXCR4 upregulation by R848 (rho = −0.5103, *p* = 0.005, Figure 2B). Among the studied conditions, CD70 expression on pDCs after mRNA-LNP, MPLA, or R848 treatment exhibited the strongest correlations with acute antibody responses.

Following univariate analysis, we performed multiple linear regression to define the clusters of activation markers within each immune cell population that exhibited the strongest association with 1-month IgG responses. Monocyte activation by Poly I:C and cDC2 activation by mRNA-LNP explained 31.04% (multiple R^2^ = 0.3104, *p* = 0.008) and 29.05% (multiple R^2^ = 0.2905, *p* = 0.012) of IgG response variance, respectively. However, R848-induced cDC2 activation (CD86, CD40, CD25) was the strongest predictor, accounting for 47.01% of variance (multiple R^2^ = 0.4701, *p* = 0.001) (Table 3).

We next conducted a multiple linear regression analysis to define the clusters of activation markers across each immune cell population that exhibited the strongest association with 1-month antibody responses. This analysis identified three models strongly correlating with anti-spike IgG levels. MPLA-induced changes in cDC2 (CD38), pDC (CD70), and NK cell (CD25, CD38) activation explained 59.9% of antibody response variance (multiple R^2^ = 0.599, *p* ≤ 0.001). The Poly I:C-induced expression of CD70 and CD80 on monocytes, along with CXCR4 on pDC, and CD38 on B cells, accounted for 60.2% of the variation in IgG levels (multiple R^2^ = 0.6024, *p* ≤ 0.001). The strongest predictor was an R848-based model incorporating cDC1 (CD80), pDC (CD70), and cDC2 (CD40, CD25, CD86) activation, which explained 68.1% of variance (multiple R^2^ = 0.681, *p* ≤ 0.001) (Table 3). These data indicate that the pre-vaccination assessment of TLR3, TLR7/8, and TLR4 engagement in innate immune cells, particularly cDCs and pDCs, correlates with the anti-spike IgG response magnitude. Notably, the models correlating with an acute antibody response (R848, Poly I:C, MPLA) differed from those correlating with reactogenicity (baseline activation, mRNA-LNP, Poly I:C). Although models utilizing Poly I:C stimulation resulted in significant correlations for both, no markers overlapped between the two.

We further analyzed the correlation of cytokine production from stimulated PBMCs with the magnitude of anti-spike IgG 1-month post-boost. None of the cytokines measured in the culture supernatants individually exhibited a statistically significant correlation with antibody levels. While cytokine production after empty LNP and MPLA stimulation exhibited trends towards inverse correlations, and mRNA-LNP stimulation showed trends toward positive correlations, these were not statistically significant (Figure 2C).

However, when we assessed associations of cytokine combinations through multiple linear regression, the mRNA-LNP-induced production of IL-12, IL-8, and TNFα explained 58.69% of the variance in acute antibody responses (multiple R^2^ = 0.5869, *p* ≤ 0.001). Similarly, the Poly I:C-induced production of GM-CSF, TNFα, and IL-6 accounted for 39.93% of this variance (multiple R^2^ = 0.3993, *p* = 0.005, Table 3). Integrating cytokine production data with surface marker expression, however, did not yield models with a better fit.

Our findings indicate that individual correlations with stimulation-induced marker expression yielded significant associations, and multiple linear regression identified strong potentially predictive models of anti-spike IgG responses 1-month post-vaccination. These immunogenicity correlates distinctly contrasted with those for reactogenicity, which were most strongly associated with the baseline activation status of PBMCs and responses to mRNA-LNPs and Poly I:C.

### 3.5. Innate Immune Responses That Correlate with Antibody Durability Differ from Those That Correlate with Reactogenicity and Antibody Levels at 1-Month Post-Vaccination

In addition to examining relationships with acute vaccine responses, we sought to identify associations between innate immune activation and anti-SARS-CoV-2 IgG magnitude 6 months post-vaccination. While 6-month IgG responses correlated with 1-month levels (Figure 3A), our analysis identified distinct innate immune patterns associated with antibody durability.

When evaluating individual phenotypic markers on immune cells at baseline, we found that baseline HLA-DR expression on pDCs was inversely correlated with the 6-month antibody response (rho = −0.3714, *p* = 0.047), while CD38 on NK cells was positively associated (rho = 0.3968, *p* = 0.033, Figure 3B). Unlike reactogenicity, no other baseline measure correlated with 6-month antibody levels.

For stimuli-induced changes, the Poly I:C upregulation of HLA-DR (rho = 0.3744, *p* = 0.045) and CXCR4 (rho = 0.397, *p* = 0.033) on monocytes positively correlated with the antibody response at 6-months post-vaccination. This finding is distinct from both reactogenicity and the 1-month IgG response. In contrast to associations with the 1-month response, B-cell activation by R848, as demonstrated by the upregulation of CD86 (rho = 0.4000, *p* = 0.032) and CD40 (rho = 0.4325, *p* = 0.019), significantly correlated with durability (Figure 3B). The NK cell expression of CD38 after mRNA-LNP treatment (rho = 0.4267, *p* = 0.021), as well as CXCR4 after MPLA stimulation (rho = 0.3847, *p* = 0.039), positively associated with durability but not with acute antibody responses. Similarly to 1-month antibody responses, the R848-induced upregulation of CD80 (rho = 0.4522, *p* = 0.014) and CD40 (rho = 0.4010, *p* = 0.311) by cDC1s exhibited positive correlations with durability.

The univariate analysis of cytokine production identified only one significant association: an inverse correlation of IL-10 production (rho = −0.436, *p* = 0.018) after Poly I:C stimulation with 6-month antibody responses (Figure 3C).

We next used multiple linear regression to identify innate activation models with enhanced associations with anti-spike IgG durability. The strongest model, defined by a baseline expression of CD40 by cDC2s, HLA-DR by pDCs, and both CD70 and CD14 by monocytes, collectively explained 55.58% of the variance in 6-month IgG levels (multiple R^2^ = 0.5558, *p* ≤ 0.001). The Poly I:C-induced activation of monocytes (CD80 and CD70) and cDC2s (CD80 and CD86) predicted 46% of durability (multiple R^2^ = 0.46, *p* = 0.004). These models suggest that both baseline and dsRNA sensor-induced innate immune activation could partially predict durability (Table 4). The inclusion of IL-10 further strengthened the Poly I:C model, enhancing the multiple R^2^ to 0.5464 (*p* ≤ 0.001) (Table 4).

Although the 6-month IgG response correlated with 1-month levels (Figure 3A), distinct patterns of innate immune activation emerged as correlates of durability. These associations included Poly I:C-induced HLA-DR upregulation in monocytes, which was not associated with the 1-month antibody response. Additionally, R848-induced costimulatory molecule expression on cDC1s correlated with both 1- and 6-month IgG responses. Conversely, baseline HLA-DR expression in pDCs and IL-10 production after the Poly I:C stimulation of PBMCs were inversely associated with durability, but not 1-month responses. While baseline expression did correlate with 6-month durability, the specific surface molecules differed from those associated with reactogenicity. Poly I:C-induced CD70 upregulation on monocytes, in combination with the activation of other innate immune cells, correlated with 1- and 6-month antibody responses. This was not seen with reactogenicity (Table 4). Unlike with reactogenicity and 1-month antibody responses, Poly I:C-induced CD80 and CD86 upregulation on cDC2s correlated with 6-month antibody titers. The predictive strength of these multiple regression models for 6-month antibody levels was not as high as for reactogenicity or 1-month antibody levels, likely due to unmeasured variables influencing host immune responses during the intervening period.

## 4. Discussion

The analysis of pre-vaccination PBMCs in a well-characterized clinical cohort of BNT162b2 vaccine recipients identified the correlates of reactogenicity and immunogenicity at 1- and 6-months post-vaccination. Individual- and population-based predictors of vaccine efficacy and adverse reactions are limited but have the potential to improve vaccine administration and boosting as well as vaccine design. By investigating the resting state of PBMCs and their responsiveness to in vitro stimulation, we fit the models of these innate immune factors to predict reactogenicity and the magnitude and durability of the IgG responses following vaccination with the BNT162b2 vaccine. The strongest predictor of reactogenicity, explaining 79% of the individual variation in symptom scores, was the resting activation state of cells from the peripheral blood. Immunogenicity at one month most strongly correlated with R848-induced activation, describing 68% of the variability in the 1-month IgG response. Lastly, 6-month antibody durability was most strongly associated with Poly I:C-induced activation, accounting for 55% of the variability in durability. Our findings offer initial insights for predictive modeling, which may be utilized to screen individuals prior to vaccination to assess their likely reactogenicity and antibody responses. Furthermore, the models identified in our analysis highlight key cell populations and signaling pathways that may be critical targets for the advancement of vaccine design, but further mechanistic analysis is required to confirm these connections.

Our investigation into the correlates of reactogenicity found that the resting state of innate immune populations, particularly pDCs, showed the strongest association with reactogenicity to the BNT162b2 vaccine. A model integrating the resting states of immune cell populations accounted for 79% of the individual variation in post-vaccination symptom scores, highlighting its potential robust predictive capacity. This strong association indicates that individuals with a lower baseline activation of immune cells prior to vaccination may be at an increased risk of experiencing substantial post-vaccination side effects. Beyond the baseline activation status of immune cells, the functional abilities of monocytes, B cells, and NK cells to respond to innate stimuli also emerged as significant predictors of the severity of symptoms upon receipt of the second BNT162b2 vaccine dose.

While reactogenicity is often linked to the production of proinflammatory cytokines shortly after vaccination, our study identified the baseline expression of activation markers as the key predictive model for adverse symptoms rather than cytokine production in response to in vitro stimulation [13,14,15]. This model was primarily driven by associations with dendritic cell resting states, aligning with findings from other vaccine studies [20]. Given their crucial roles in antigen presentation and immune activation, these innate immune cells are key populations investigated in vaccine immune response studies. Although we did not explore the mechanistic basis of the observed correlations, it is interesting to postulate that the lower baseline activation states of dendritic cells necessitate a greater magnitude of the inflammatory response for their full activation. This increased inflammatory response may lead to a more intense reaction and, consequently, greater reactogenicity. Moreover, using the baseline activation state as a marker for reactogenicity provides a less labor-intensive predictive approach.

To further define pre-vaccination correlates, we also investigated the activation profiles of early-responding immune cells like monocytes, B cells, and NK cells to various innate stimuli in relation to post-vaccine symptoms. Our findings identified monocyte activation in response to mRNA-LNPs to be potentially predictive of reactogenicity, accounting for approximately 56% of symptom score variability. This association between the monocyte response to vaccination and reactogenicity has previously been identified through an assessment of PBMCs after influenza vaccination [20]. Additionally, others have demonstrated associations between symptoms after BNT162b2 vaccination and both NK cells and DC3s [12]. Although we did not investigate this DC population, our study and those of others have shown NK cell associations with reactogenicity [30]. We identified that the stimulation-induced upregulation of CD38 on NK cells correlates significantly with the magnitude of the symptom score post-boost, suggesting a multifaceted role for NK cells in vaccine responses. Our study adds to the existing literature by demonstrating that innate immune associations with reactogenicity can be observed in vitro using pre-vaccination PBMCs.

In contrast to other studies evaluating cytokines in the peripheral blood post-vaccination that found a correlation between systemic post-vaccination symptoms and heightened production, our in vitro analysis did not identify strong cytokine–reactogenicity correlations. Studies investigating the response to various vaccine platforms, including a hepatitis B subunit vaccine, identified cytokines such as IFN-γ, IL-6, IL-1β, and TNFα in the peripheral blood post-vaccination [13,14,15]. Relative levels of these cytokines have been shown to predict adverse events [15]. For BNT162b2, the magnitude of IFN-γ, CCL20, and CXCL9-11 present in the serum within two days post-vaccination correlated with reactogenicity [11]. Our panel did not include CCL20, CXCL9, and CCL11, and it is possible that the inclusion of additional cytokines, such as these, may reveal additional predictors of reactogenicity. However, it remains uncertain if including additional cytokines would yield more robust predictors, especially since IFN-γ, a cytokine previously linked to reactogenicity, did not show associations in our analysis. Additionally, standardizing symptom evaluation and scoring, including separate analyses for local and systemic symptoms, will be crucial for future analysis to identify more refined predictive markers. Consistent with this, a prior report correlates systemic symptoms with the serum levels of IFN-γ, IL-6, and IP-10 following BNT162b2 vaccination [12].

In addition to investigating immune predictors of reactogenicity, we sought to identify pre-vaccination innate immune correlations with immunogenicity in response to BNT162b2 vaccination. Within our study, the magnitude of reactogenicity and immunogenicity did not correlate. While innate immune responses play a critical role in both reactogenicity and immunogenicity, evidence for direct associations between the two is inconsistent. An analysis of the complete PASS cohort did not identify a significant relationship between BNT162b2-induced side effects and peak antibody titers [7], aligning with findings from other studies [38]. Furthermore, reactogenicity and immunogenicity are not always linked across different vaccine platforms [13,16,19]. However, some studies have reported a correlation between reactogenicity and higher IgG levels following BNT162b2 vaccination [39,40], with evidence suggesting that this association may be sex-dependent, as it is observed primarily in males [41]. As the PASS cohort is predominantly female, this may explain the discrepancy. However, further investigation into the nuanced relationship between reactogenicity and antibody responses is warranted. Thus, we investigated distinct immune associations with the magnitude of the anti-spike IgG response utilizing our predominantly female cohort.

The immune factors correlating with acute (1-month) antibody responses did not substantially overlap with those that correlated with vaccine durability (6-month antibody responses), despite a positive correlation between 1- and 6-month antibody levels. For acute response, the level of pDC activation in response to TLR agonists, empty LNPs, and mRNA-LNPs inversely correlated with anti-spike IgG levels. The best predictive model for acute antibody responses was the R848-induced activation of dendritic cells, which could account for 68% of the variability in 1-month post-vaccination IgG levels. In contrast, positive associations were observed between innate immune responsiveness and antibody durability. For antibody durability, the strongest model, explaining 55% of the variability in 6-month post-vaccination IgG levels, was based on the Poly I:C-induced activation of monocytes and cDC2s, combined with an inverse association with IL-10 production from PBMCs. The inverse correlation seen with IL-10 is consistent with other studies demonstrating inverse associations with antibody responsiveness [42,43]. Overall, innate immune measures demonstrated a robust predictive capacity for reactogenicity and acute antibody responses and a more moderate predictive capacity for antibody durability.

For the 1-month response to vaccination, we identified innate cell responsiveness to R848 to be the most robust predictive model. This model, incorporating cDC1, pDC, and cDC2 activation in response to R848 stimulation, explained 68% of the variance, making it the strongest predictive model. Additionally, MPLA-induced changes in cDC2, pDC, and NK activation explained 60% of the antibody response variance. These data collectively demonstrate associations between dendritic cell responses and the development of antibody responses to BNT162b2 vaccination, supporting the central role of DCs in antigen presentation and immune activation. Furthermore, our findings are consistent with studies showing the upregulation of antigen presentation genes correlating with acute antibody responses to influenza vaccination [26]. This suggests that in vitro assessments of pre-vaccination innate responses, particularly classical antigen-presenting cell responses, may be valuable for predicting acute antibody magnitude.

Our data also indicated an inverse association with pDC activation and acute antibody magnitude. This finding is perhaps unsurprising given that pDCs are potent producers of IFN, which have been previously shown to have inverse correlations with 1-month responses [22]. However, we found no significant associations between individual IFNs or combinations of IFNs and 1-month antibody responses. The role of IFN production in vaccine responses is complex, as it has been shown to associate with the adaptive response in some platforms, such as inactivated vaccines [23,26,44], but not all [22].

Additionally, an investigation of monocytes identified a weak association between 1-month responses and their stimulation-induced activation. Only 31% of the variation in 1-month levels was attributable to monocyte activation. Other studies have used transcriptional signatures post-BNT162b2 vaccination, the upregulation of the CD86 costimulatory molecule in response to TLR agonists, or adenovirus vaccination to correlate robust antibody responses against SARS-CoV-2 with monocyte activation [10,42]. However, conflicting evidence exists regarding the correlation between monocyte activation and the IgG response post-influenza vaccination [18,20]. Given the inconsistencies across studies, it is clear that while monocyte activation can be linked to antibody responses in certain vaccine contexts, other factors are at play. These likely include the adjuvants used in different vaccines and host factors such as age and sex. Unfortunately, we were unable to assess age and sex as potential confounding variables, as our study cohort was primarily comprising young and female participants.

Overall, this study identified that the baseline expression of activation markers on innate immune cells has a strong inverse correlation with reactogenicity. This indicates that individuals with greater costimulatory molecule expression are less prone to BNT162b2 vaccine adverse effects. Our models evaluating innate immune cell activation in response to mRNA-LNPs, MPLA, and Poly I:C, while slightly less predictive, were positively correlated with vaccine-mediated side effects. From these models, one can hypothesize that both RNA and LNP sensing contribute to reactogenicity. Similarly, R848, Poly I:C, and MPLA models strongly correlated with acute antibody responses, with R848-induced activation being the strongest predictor. This may indicate a role for RNA sensing in acute antibody responses. However, direct causation requires further investigation. For antibody durability, baseline resting surface marker expression and Poly I:C activation models showed moderate predictive capability, emphasizing the importance of both innate immune resting states and activation responses.

Although prior studies have evaluated correlates of immunogenicity and reactogenicity, reliable predictors of these outcomes prior to vaccine administration remain elusive. The ability to predict the immunogenicity and reactogenicity of vaccines before administration may enable the earlier assessment of these critical factors. Additionally, the identification of associations with key cell populations and signaling pathways may enhance vaccine design by facilitating the development of mechanistic hypotheses. Currently, our study predominantly evaluates the response of young females and may be most predictive for this demographic. Therefore, further investigation is required to understand how these findings apply to males, children, and the elderly. Nevertheless, this study identified models that demonstrate high potential predictive value for reactogenicity, as well as antibody response 1- and 6-months post-vaccination, offering valuable insights for future vaccine development and personalized vaccination strategies.

## Figures and Tables

**Figure 1 vaccines-13-00718-f001:**
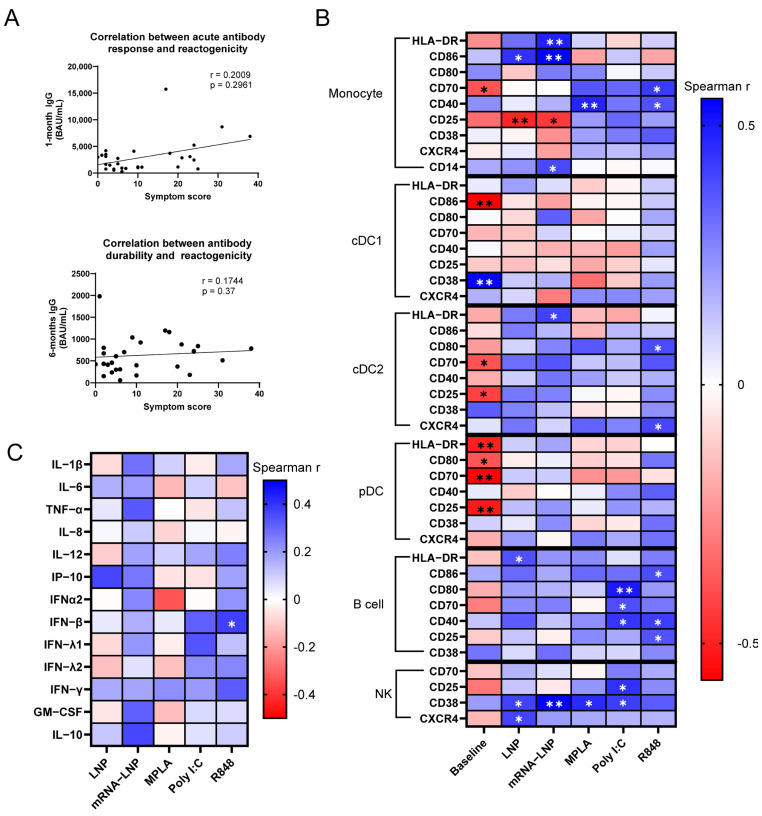
Cellular and cytokine correlates of reactogenicity to BNT162b2 vaccination. (**A**) Correlation of symptom score with anti-spike IgG titers at 1- or 6-months post-vaccination in the PASS cohort. (**B**) Correlation of symptom scores with baseline surface marker expression of unstimulated pre-vaccination PBMCs (MFI) or changes in surface marker expression following 24 h stimulation (ΔMFI). Pre-vaccination PBMCs were stimulated with empty LNPs, mRNA-LNPs, MPLA (TLR4), Poly I:C (TLR3), and R848 (TLR7/8), or left unstimulated. (**C**) Correlation of symptom scores with cytokine production in PBMC supernatants following 24 h stimulation. (n = 29 for all panels) Data are represented as individual data points (**A**) or as Spearman rho coefficients (**B**,**C**). Significance was assessed using Spearman correlations. * *p* ≤ 0.05 and ** *p* ≤ 0.01.

**Figure 2 vaccines-13-00718-f002:**
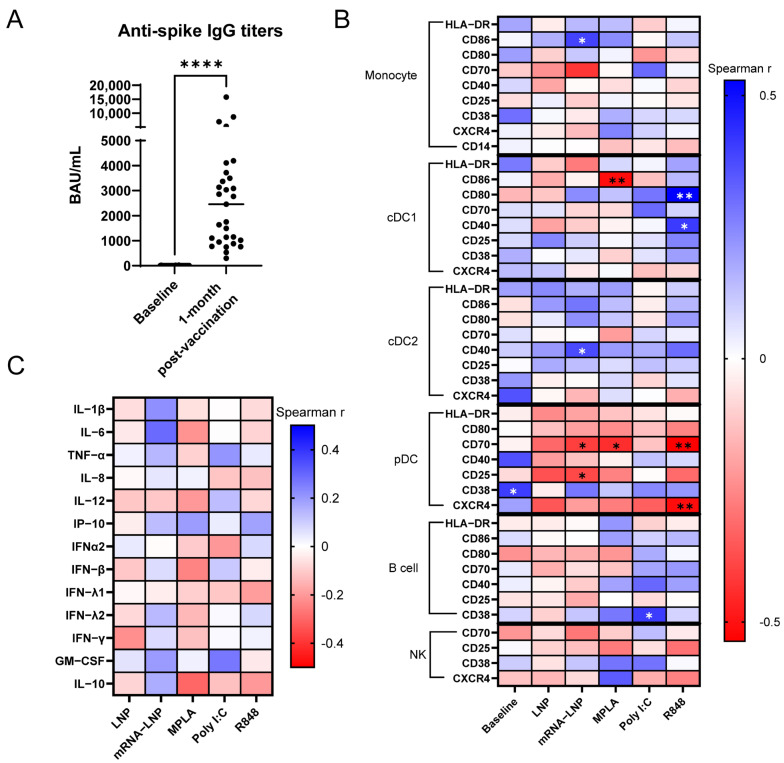
Cellular and cytokine correlates of anti-spike IgG levels 1-month post-BNT162b2 vaccination. (**A**) Increase in anti-spike IgG titers 1-month post-vaccination compared to the baseline enrollment pre-vaccination time point in the PASS cohort. (**B**) Correlation of anti-spike IgG titers 1-month post-vaccination with baseline surface marker expression of unstimulated pre-vaccination PBMCs immediately post-thaw (MFI) or changes in surface marker expression following 24 h stimulation (ΔMFI). These pre-vaccination PBMCs were stimulated with empty LNPs, mRNA-LNPs, MPLA (TLR4), Poly I:C (TLR3), and R848 (TLR7/8), or left unstimulated. (**C**) Correlation of anti-spike IgG titers 1-month post-vaccination with cytokine production in PBMC supernatants following 24 h stimulation. (n = 29 for all panels) Data are represented as individual data points with the median value indicated (**A**) or as Spearman rho coefficients (**B**,**C**). Significance was assessed using Spearman correlations (**B**,**C**). * *p* ≤ 0.05, ** *p* ≤ 0.01, and **** *p* ≤ 0.0001.

**Figure 3 vaccines-13-00718-f003:**
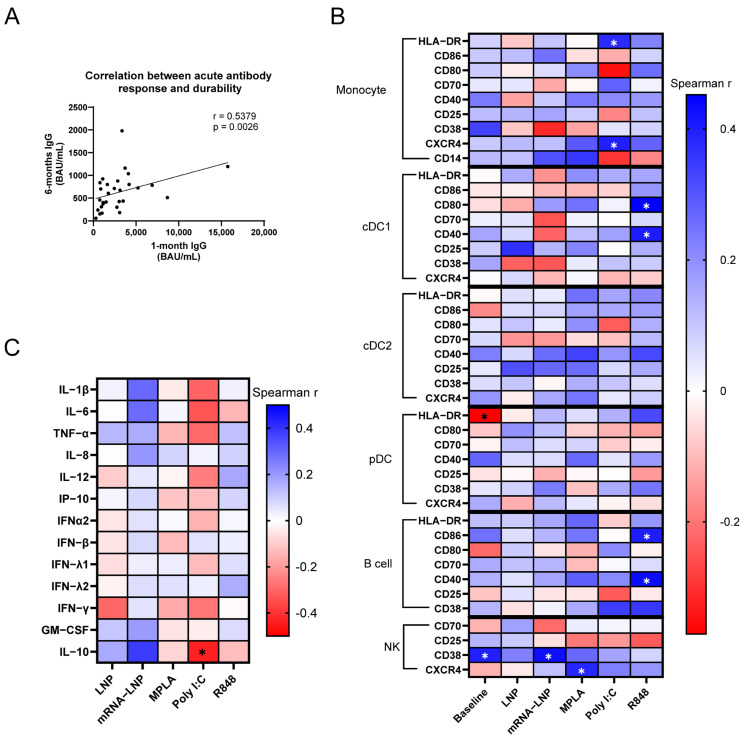
Cellular and cytokine correlates of anti-spike IgG levels 6-months post-BNT162b2 vaccination. (**A**) Correlation of anti-spike IgG titers 6-months post-vaccination with acute 1-month titers in the PASS cohort. (**B**) Correlation of anti-spike IgG titers 6-months post-vaccination with baseline surface marker expression of unstimulated pre-vaccination PBMCs immediately post-thaw (MFI) or changes in surface marker expression following 24 h stimulation (ΔMFI). These pre-vaccination PBMCs were stimulated with empty LNPs, mRNA-LNPs, MPLA (TLR4), Poly I:C (TLR3), and R848 (TLR7/8), or left unstimulated. (**C**) Correlation of anti-spike IgG titers 6 months post-vaccination with cytokine production in PBMC supernatants following 24 h stimulation. (n = 29 for all panels) Data are represented as individual data points (**A**) or as Spearman rho coefficients (**B**,**C**). Significance was assessed using Spearman correlations. **p* ≤ 0.05.

**Table 1 vaccines-13-00718-t001:** Cohort demographics and response to BNT162b2.

		Cohort n = 29
	Sex	
Demographics	Female	24
	Male	5
	**Ethnicity**	
	Non-Hispanic	26
	Hispanic	3
	Non-specified	0
	**Race**	
	White	19
	Black	2
	Asian	5
	Multiracial	2
	Not reported	1
	**Age**	
	Median age (IQR)	41.0 (35.0–47.0)
1-month post-vaccination	**Anti-spike IgG response (post-second vaccination)**	
	Median titer (range) (BAU/mL)	2456.1 (296.8–15,746.4)
	**Days post-second vaccination**	
	Median days (range)	29 (21–44)
6-months post-vaccination	**Anti-spike IgG response (post-second vaccination)**	
	Median titer (range) (BAU/mL)	605.6 (57.9–1980.0)
	**Days post-second vaccination**	
	Median days (range)	183 (166–200)
Reactogenicity	**Symptom score (post-second vaccination)**	
	Median symptom score (range)	7 (0–38)

**Table 2 vaccines-13-00718-t002:** Identification of immune correlates of adverse reactions with multiple linear regression modeling.

	Reactogenicity				
Stimulation	Cell Subset	Marker	Multiple R-Squared	Adjusted R-Squared	*p*-Value
Baseline	pDC	CD70, HLA-DR, CD25	0.4795	0.4171	<0.001
	Monocyte	CD70, CD40	0.2877	0.2329	0.012
	cDC2	CD70, CD38, HLA-DR, CXCR4	0.4846	0.3987	0.002
	cDC1	CD86, CD25, CXCR4	0.4107	0.34	0.004
	B cell	CD70, CD86, HLA-DR, CD40, CD38	0.5224	0.4185	0.003
	Across cell subsets	pDC [CD70, CD25] B cell [CD40, HLA-DR, CD38] cDC1 [CD86] cDC2 [HLA-DR] Monocyte [CXCR4]	0.7895	0.6897	<0.001
Empty LNP	Across cell subsets	NK [CD38] cDC2 [CD86]	0.2862	0.2313	0.012
mRNA-LNP	NK	CD38, CD25	0.3233	0.2713	0.006
	Monocyte	CD86, CD14, HLA-DR, CD40, CD80	0.567	0.4728	0.001
	Across cell subsets	NK [CD38] cDC1 [CD80] Monocyte [CD14, CD86]	0.5446	0.4687	<0.001
	Cytokines	IP-10, IL-8, IFN-γ	0.4834	0.4214	<0.001
	Cytokines and across cell subsets	IP-10 NK [CD38] Monocyte [CD86]	0.5875	0.538	<0.001
MPLA	NK	CD38, CXCR4	0.206	0.145	0.050
	Monocyte	CD70, CD40, HLA-DR, CD25	0.3959	0.2953	0.014
	cDC2	CD80, CD86, HLA-DR	0.3174	0.2354	0.021
	Across cell subsets	Monocyte [CD70, HLA-DR] cDC2 [CD80] pDC [CD86]	0.5078	0.4258	0.001
Poly I:C	NK	CD38, CD70	0.3582	0.3088	0.003
	B cell	CD80, CD40	0.4332	0.3896	<0.001
	Across cell subsets	NK [CD38, CD70] B cell [CD80, CD40]	0.5753	0.4829	<0.001
	Cytokines	IL-8, GM-CSF	0.4504	0.3845	0.002
	Cytokines and across cell subsets	IFN-β, IL-8, GM-CSF B cell [CD40, CD80] NK [CD70]	0.6347	0.535	<0.001
R848	NK	CD70, CD38	0.3448	0.2944	0.004
	cDC2	CD80, CD70	0.1383	0.1064	0.047
	Across cell subsets	Monocyte [CD70] cDC1 [CD80]	0.2902	0.2356	0.012

Multiple linear regression was performed in four ways to assess models for correlations with symptom scores: (1) combination of surface markers within cell populations, (2) surface marker expression across cell populations, (3) production of combinations of cytokines, and (4) cytokine production coupled with surface marker expression across cell populations. These analyses were performed separately for each condition: baseline state, empty LNP, mRNA-LNP, Poly I:C (TLR3), R848 (TLR7/8), or MPLA (TLR4). Statistically significant analyses were reported.

**Table 3 vaccines-13-00718-t003:** Identification of immune correlates of acute anti-spike antibody responses with multiple linear regression modeling.

1-Month IgG					
Stimulation	Cell Subset	Markers	Multiple R-Squared	Adjusted R-Squared	*p*-Value
Baseline	cDC2	CD38, CXCR4, CD86	0.4029	0.3312	0.004
Empty LNP	cDC1	CD25, CD40	0.3345	0.2834	0.005
	Across cell subsets	cDC1 [CD25, CD40] cDC2 [CD25]	0.3836	0.3096	0.006
mRNA-LNP	cDC2	CD40, CD86	0.2905	0.2359	0.012
	Cytokines	IL-12, IL-8, TNFɑ	0.5869	0.5373	<0.001
MPLA	pDC	CD38, CD70	0.2946	0.2403	0.011
	NK	CD38, CD25	0.3317	0.2803	0.005
	Across cell subsets	cDC2 [CD38] pDC [CD70] NK [CD25, CD38]	0.599	0.5322	<0.001
Poly I:C	pDC	CD38, CXCR4, CD40	0.3343	0.2544	0.016
	Monocyte	CD70, CD80	0.3104	0.2573	0.008
	cDC1	CD70, CD80	0.1615	0.1305	0.031
	B cell	CD38, CD80	0.2511	0.1935	0.023
	Across cell subsets	Monocyte [CD70, CD80] pDC [CXCR4] B cell [CD38]	0.6024	0.5362	<0.001
	Cytokines	GM-CSF, TNFɑ, IL-6	0.3993	0.3273	0.005
R848	pDC	CD38, CD70	0.3938	0.3471	0.001
	NK	CXCR4, CD38, CD25	0.3196	0.2379	0.020
	cDC2	CD86, CD40, CD25	0.4701	0.4066	0.001
	Across cell subsets	cDC1 [CD80] pDC [CD70] cDC2 [CD40, CD25, CD86]	0.681	0.6116	<0.001

Multiple linear regression was performed in four ways to assess models for correlations with a 1-month IgG titer: (1) combination of surface markers within cell populations, (2) surface marker expression across cell populations, (3) production of combinations of cytokines, and (4) cytokine production coupled with surface marker expression across cell populations. These analyses were performed separately for each condition: baseline state, empty LNPs, mRNA-LNPs, Poly I:C (TLR3), R848 (TLR7/8), or MPLA (TLR4). Statistically significant analyses are reported.

**Table 4 vaccines-13-00718-t004:** Identification of immune correlates of anti-spike antibody durability with multiple linear regression modeling.

6-Month IgG					
Stimulation	Cell Subset	Marker	Multiple R-Squared	Adjusted R-Squared	*p*-Value
Baseline	pDC	CD40, HLA-DR	0.2662	0.2098	0.018
	Monocyte	CD38, CD70, CD14, CXCR4	0.3533	0.2455	0.028
	cDC2	CD40, CD25	0.323	0.2709	0.006
	Across cell subsets	cDC2 [CD40] pDC [HLA-DR] Monocyte [CD70, CD14]	0.5558	0.4817	<0.001
Empty LNP	cDC1	CD25, CD80, CD38	0.2965	0.2121	0.030
MPLA	Monocyte	CD14, CXCR4	0.2137	0.1532	0.044
	B cell	HLA-DR, CD25	0.18	0.1169	0.076
Poly I:C	Monocyte	CD80, CD70	0.2743	0.2185	0.015
	cDC2	CD80, CD86	0.3254	0.2735	0.006
	Across cell subsets	Monocyte [CD80, CD70] cDC2 [CD80, CD86]	0.46	0.37	0.004
	Cytokines and across cell subsets	IL-10 cDC2 [CD80, CD86] Monocyte [CD70]	0.5464	0.4708	<0.001
R848	B cell	CD40, CD25	0.3481	0.298	0.004

Multiple linear regression was performed in four ways to assess models for correlation with 6-month IgG titer: (1) combination of surface markers within cell populations, (2) surface marker expression across cell populations, (3) production of combinations of cytokines, and (4) cytokine production coupled with surface marker expression across cell populations. These analyses were performed separately for each condition: baseline state, empty LNPs, mRNA-LNPs, Poly I:C (TLR3), R848 (TLR7/8), or MPLA (TLR4). Statistically significant analyses are reported.

## Data Availability

Data are available from the corresponding author upon request.

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
