# Peer review of "Pre-Vaccination Immune Profiles and Responsiveness to Innate Stimuli Predict Reactogenicity and Antibody Magnitude Following mRNA Vaccination"

_vaccines, 2025, doi:10.3390/vaccines13070718_

Round 1
Reviewer 1 Report
Comments and Suggestions for Authors
This study presents an investigation into the potential of pre-vaccination immune profiling to predict the reactogenicity and immunogenicity following BNT162b2 mRNA vaccination. The authors utilize a comprehensive approach, combining in vitro stimulation of PBMCs with mRNA-LNP, LNP, and TLR agonists, followed by multiparameter flow cytometry and cytokine profiling. The findings contribute valuable insights into the role of innate immune activation in shaping vaccine responses and adverse effects. The manuscript is well-written, and the statistical analyses are robust. However, some aspects require clarification and further discussion.
1) The cohort is predominantly 82.7% female with a specific age range (median 41 years). Given known sex- and age-related differences in immune responses, how might this bias affect the generalizability of the findings?
2) The doses for LNP/mRNA-LNP and TLR agonists were selected based on CD86 upregulation in monocytes. Were any toxicity or viability controls performed at these doses?
3) The cytokine panel did not include CCL20, CXCL9, and CXCL11, which have been previously linked to reactogenicity. Would expanding the panel enhance the findings?
Reviewer 2 Report
Comments and Suggestions for Authors
Overall comments
Vaccinees exhibit a wide range of range of responses after vaccination. Some vaccinees produce high or low titers of antibodies against the vaccine immunogen. The antibody responses of some vaccinees are durable, while the antibody response of other are not. Some vaccinees experience a highly reactogenic response, while others experience less reactogenicity. The causes of these differences in vaccinee responses are not clear. This manuscript describes lymphocyte characteristics prior to vaccination with the BioNTech/Pfizer BNT162b2 mRNA vaccine (BNT162b2), correlating those characteristics with anti-spike antibody responses at 1 and 6 months following vaccination with BNT162b2, and with reactogenicity scores. The manuscript also describes correlations between antibody responses and reactogenicity scores. This is valuable data overall. However, there are some curious omissions and the writing and presentation of the data detract from its potential impact.
Overall, the main conclusions of the manuscript, that there are some characteristics of resting PBMCs that correlate with anti-spike antibody production and durability of the antibody response and with reactogenicity are valuable.
Major
The paper is written in a very dense and difficult to understand way.
The figure labels and captions are difficult to understand and presented in a way that obscures the data (more below).
One missing category of data missing from this manuscript is neutralization. The paper describes anti-spike antibody binding data, but presents no neutralization data. Sometimes antigen binding and neutralization are well-correlated, but sometimes not. Neutralization is what counts clinically.
Another missing category of data missing from this manuscript is data concerning cellular immune responses. Both humoral and CMI play a role in protection. PBMCs were collected from the study participants prior to vaccination, since data obtained using those cells constitute the heart of this paper. Were cells not collected after vaccination? Were there correlations in the CMI responses to the initial data?
Among the entry/research subject sample selection criteria (line 130) listed was “…had anti-spike binding IgG antibody serum levels in the upper or lower quintile 1 month after the 2nd vaccination.” What is the justication for this limitation? Some of the data (e.g. Figure 1A) are borderline significant, and might indeed be significant if more data were available, or may be compromised by one or a few data points wildly outside the rest. In fact, looking at the data for the 6-month IgG result (1A, lower panel), it is not clear that there are two obviously separable populations, a high quintile and low quintile. Excluding the middle 60% of potential research subjects could substantially compromise the results and their conclusions.
Minor
The list of abbreviations at the end of the manuscript is incomplete
Introduction
The Introduction provides adequate background information.
Methods
The methods section provides an adequate description of the methods used.
How close are the LNPs used in this study to stimulate the PBMCs to the LNPs of BNT162b2?
Results
The results in general appear sound. Some concerns, particularly with respect to study design, sample selection, and data presentation as discussed above.
Conclusions/Discussion
Some of the conclusions are fairly far-reaching and lack specifics. For example, “Our findings begin to bridge the gap in predictive modeling to help inform personalized vaccine strategies and advancement of vaccine design, but further mechanistic analysis is required to confirm these results.” (Line 597). How exactly will the data actually “inform personalized vaccine design”.
The phrase “low activation states of dendritic cells associate with heightened reactogenicity because a greater inflammatory response may be required for their activation” (line 620) doesn’t really make much sense and reads like an example of backwards reasoning.
The data presented in this manuscript are really correlations. Sometimes, the manuscript seems to imply that the data have mechanistic implications, but the data as presented doesn’t really speak to mechanisms.
The discussion should include a mention of the observations that LNPs themse
Figures and Tables
In all the figures, the subpanel call-outs (e.g. A, B…) are placed after the descriptive text for each subpanel. This is confusing, particularly for the A subpanels because it is hard to know where the initial description/title of the figure as a whole ends and the description of the individual subpanels begins. This is really contrary to common practice. In addition, the legend for Figure 2 does not have any call-out for subpanel A.
In subpanel C of Figure 1, somewhere it should say “cytokine production” or “cytokine production after cell treatment” or something equivalent.
Instead of just “LNP” it would be clearer to say “Empty LNP” to more clearly distinguish between those particles and the particles that carry the nRNAs.
“Unstimulated” would be a clearer way to describe the cells as opposed to “Baseline”. Baseline can be confused with cells obtained prior to vaccination vs. after subsequent sampling. In the results for the stimulated cells, having a bar across the bottom of the figure with text something like “stimulated with” would also make these figures much clearer.
Splitting tables across pages makes the manuscript harder to read.
Comments on the Quality of English LanguageThe English language is OK, but the presentation is difficult. It would be helpful if the authors found someone unaffiliated with the work who could give them a very careful and critical read, and then revise the manuscript accordingly.
Reviewer 3 Report
Comments and Suggestions for Authors
Here, Zelkoski et al present a study using cytokine analysis and spectral flow cytometry to correlate innate immune responses to the magnitude of antibody responses and reactogenicity of the BNT162b2 mRNA vaccination. The authors have a great control of pre-vaccination PBMCs which they can use for stimulation assays and multiparameter flow cytometry within a well-characterised clinical cohort, offering novel insights into predictive immune signatures. The study is easy to understand and methodologically sound. It also fits nicely into a gap in our understanding of individualised vaccine responses.
The work is well controlled and the spectral flow is a more novel way of analysing these markers. I’m quite happy with the statistics and the study design.
A few things that need changing:
- It is worth noting that vaccines also appear to protect against long term symptoms of COVID-19, and if anything are now the more important aspect of vaccination: (DOIs: 10.1038/s41541-022-00526-5, 10.1093/cid/ciac630, 10.1093/ofid/ofac464, 10.1038/s41591-022-01840-0)
- I would be tempted to highlight that predominantly young and female nature of the cohort as a limitation which might not apply to the population as a whole.
- The symptoms score questionnaire mentioned throughout needs to be included so readers can evaluate the usefulness of these scores.
- The race numbers only add to 28 instead of 29.
- Figure 2A would be strengthened with baseline levels from pre-vaccination to show that donors increased after vaccination, and also would show if the magnitude of the increase is important (rather an absolute levels)
- I think it is a little unclear how these findings help vaccine design. Is the idea to screen people beforehand for certain types of vaccines, or to design vaccines to target particular pathways better?
Reviewer 4 Report
Comments and Suggestions for Authors
This study initially investigates and analyzes the possible factors that potentially affect the antibody titers elicited by vaccinations. In the future, further research is required to conduct a rigorous and comprehensive analysis of multiple variables in order to draw reliable conclusions.
Vaccines exhibit correlations with certain factors within specific vaccination populations to some degree, such as the strength of the innate immune responses prior to vaccination, which is the focus of this article. Nonetheless, the immunogenicity of vaccines and their effects on individuals remain a highly intricate issue influenced by various factors, including population genetic characteristics and the rearrangement effectiveness of V(D)J in heavy and light chain genes. The necessity, feasibility and significance of conducting research from these aspects warrant detailed elaboration in the discussion section.
In addition, the first and second paragraphs of the discussion section should present examples of cutting-edge research, compare and analyze correlates to reactogenicity, or address whether the author was among the pioneers in conducting these analytical studies.
Minor,
There are double C in the panel of Figure 2.
The first sentence of the Abstract is overly general and could be refined for specificity. In line 68, clarification is needed regarding whether the reduction in CD4 T cells is associated with an increase or decrease in antibody levels. On page 79, a comma should be added after "blood of adults" for improved readability. In line 181, the numeral "2" for carbon dioxide should be formatted as a subscript. Additionally, there are numerous similar issues with subscripts throughout the Method section, which requires careful attention and correction.
Comments on the Quality of English Language
Good quality.
Round 2
Reviewer 2 Report
Comments and Suggestions for Authors
The authors have made reasonable responses to the review comments.
Author Response
Comment 1: The authors have made reasonable responses to the review comments.
Response 1: Thank you for the thorough review of our work. We appreciate the opportunity to improve our manuscript.